# Cancer Malignancy Is Correlated with Upregulation of PCYT2-Mediated Glycerol Phosphate Modification of α-Dystroglycan

**DOI:** 10.3390/ijms23126662

**Published:** 2022-06-15

**Authors:** Fumiko Umezawa, Makoto Natsume, Shigeki Fukusada, Kazuki Nakajima, Fumiya Yamasaki, Hiroto Kawashima, Chu-Wei Kuo, Kay-Hooi Khoo, Takaya Shimura, Hirokazu Yagi, Koichi Kato

**Affiliations:** 1Exploratory Research Center on Life and Living Systems (ExCELLS), National Institutes of Natural Sciences, 5-1 Higashiyama, Myodaiji, Okazaki 444-8787, Japan; c212803@ed.nagoya-cu.ac.jp (F.U.); kkhoo@gate.sinica.edu.tw (K.-H.K.); 2Institute for Molecular Science (IMS), National Institutes of Natural Sciences, 5-1 Higashiyama, Myodaiji, Okazaki 444-8787, Japan; 3Graduate School of Pharmaceutical Sciences, Nagoya City University, 3-1 Tanabe-dori, Mizuho-ku, Nagoya 467-8603, Japan; 4Graduate School of Medical Sciences, Nagoya City University, 1 Kawasumi, Mizuho-cho, Mizuho-ku, Nagoya 467-8601, Japan; makoto04130510@gmail.com (M.N.); shigeki.f@hotmail.co.jp (S.F.); tshimura@med.nagoya-cu.ac.jp (T.S.); 5Center for Joint Research Facilities Support, Research Promotion and Support Headquarters, Fujita Health University, Toyoake 470-1192, Japan; knakaji@gifu-u.ac.jp; 6Institute for Glyco-core Research (iGCORE), Gifu University, 1-1 Yanagido, Gifu 501-1193, Japan; 7Graduate School of Pharmaceutical Sciences, Chiba University, 1-8-1 Inohana, Chuo-ku, Chiba 260-0856, Japan; axxa4826@chiba-u.jp (F.Y.); h-kawashima@chiba-u.jp (H.K.); 8Institute of Biological Chemistry, Academia Sinica, 128 Sec. 2, Academia Road, Nankang, Taipei 115, Taiwan; kuocw@gate.sinica.edu.tw

**Keywords:** cancer malignancy, CDP-glycerol, α-dystroglycan, glycerol phosphate modification, matriglycan, PCYT2

## Abstract

The dystrophin–glycoprotein complex connects the cytoskeleton with base membrane components such as laminin through unique O-glycans displayed on α-dystroglycan (α-DG). Genetic impairment of elongation of these glycans causes congenital muscular dystrophies. We previously identified that glycerol phosphate (GroP) can cap the core part of the α-DG O-glycans and terminate their further elongation. This study examined the possible roles of the GroP modification in cancer malignancy, focusing on colorectal cancer. We found that the GroP modification critically depends on PCYT2, which serves as cytidine 5′-diphosphate-glycerol (CDP-Gro) synthase. Furthermore, we identified a significant positive correlation between cancer progression and GroP modification, which also correlated positively with PCYT2 expression. Moreover, we demonstrate that GroP modification promotes the migration of cancer cells. Based on these findings, we propose that the GroP modification by PCYT2 disrupts the glycan-mediated cell adhesion to the extracellular matrix and thereby enhances cancer metastasis. Thus, the present study suggests the possibility of novel approaches for cancer treatment by targeting the PCYT2-mediated GroP modification.

## 1. Introduction

Epithelial cells interact with the basement membrane, a sheet-like type of extracellular matrix (ECM), through cell adhesion molecules, which regulate cellular growth, motility, and differentiation by integrating signals from ECM or soluble factors [1,2]. One of the most extensively studied cell adhesion molecules is α-dystroglycan (α-DG), which is displayed on epithelial cell surfaces as a component of the dystrophin–glycoprotein complex [3,4]. This complex connects the cytoskeleton with basement membrane components such as laminin, agrin, and perlecan and plays an essential role in epithelium development and tissue organization. The connection critically depends on unique O-glycans, termed matriglycans, attached to the N-terminal region of the mucin-like domain of α-DG [5,6,7].

The outer regions of matriglycans consist of α1-3Xylβ1-3GlcA repeat sequences formed by a dual functional glycosyltransferase LARGE and are responsible for the interactions with the base membrane components [8]. The inner part of matriglycan is constituted from the reducing terminal 6-phosphorylated core M3 structure, i.e., GalNAcβ1-3GlcNAcβ1-4Man-6-phosphate, and a tandem ribitol-5-phosphate (Rbo5P) structure followed by a β1-4Xylβ1-4GlcA disaccharide. Besides LARGE, at least nine different glycosyltransferases are involved in forming the matriglycans [9,10,11,12,13,14,15,16]. Mutational dysfunction of either of these enzymes impairs the formation of matriglycans, disrupting the α-DG–ECM interactions, causing congenital muscular dystrophies (CMD) designated as α-dystroglycanopathies. Among these enzymes, fukutin (FKTN) and fukutin-related protein (FKRP) catalyze the first and second steps of Rbo5P transfer, respectively. This donor substrate is synthesized by cytidine 5′-diphosphate (CDP)-L-ribitol pyrophosphorylase A (CRPPA/ISPD) [15,17,18,19].

We recently found that glycerol phosphate (GroP) modifies the 6-phosphorylated core M3 structure without further elongation [20]. The dual action of FKTN catalyzes this novel modification and, therefore, competes for the first Rbo5P conjugation [20,21]. Furthermore, the GroP modification is enhanced by elevation of intracellular CDP-Gro level, which can be artificially realized by overexpression of TagD, a CDP-Gro synthase from Gram-positive bacteria [22,23], resulting in a concomitant decrease of the matriglycans [24]. These findings raise the possibility that GroP serves as a terminator of the matriglycan elongation. However, the physiological and pathological functions of this modification remain largely unknown.

Mutations in causative genes such as *LARGE* and *FKRP* often result in defect or decreased expression of matriglycans [9,10,11,12,13,14,15,16]. On the other hand, matriglycans have been shown to be reduced or abolished in primary human cancer cells and cell lines, including prostate, breast, and colorectal cancers [25,26,27,28,29]. Interestingly, the defects in the matriglycans of α-DG are correlated with poor prognosis [30,31]. Since overexpression of LARGE significantly inhibits the cancer cell migration ability [32,33] and low expression of B4GAT1 results in increased cancer cell invasion, altered glycosylation of α-DG is considered to be associated with cancer development and progression [34]. However, the regulation mechanisms of matriglycan formation in human cancers remain to be uncovered.

In this study, we attempt to determine the possible roles of the GroP modification in cancer malignancy. We reveal that the GroP modification was promoted in several cancer tissues and address the molecular mechanism and pathological significance of this modification.

## 2. Results

### 2.1. GroP Modification Is Enhanced in Cancer Tissues

We previously established a monoclonal antibody DG2, reactive with GroP-modified α-DG [24]. Using this antibody, we probed the GroP modification in various normal and cancerous tissues. Several cancerous tissues, including the bladder, uterus, ovary, and colon, tested positive for immunostaining compared to their normal counterparts (Appendix A). Focusing on colorectal cancer, we investigated a relationship between degrees of GroP modification and malignancy (Figure 1). GroP expression in human colorectal cancer (CRC) tissues was analyzed using immunohistochemistry with DG2 for primary tumor tissues with surgical resection. Representative images of DG2 staining are shown in Figure 1A. The 100 CRC patients consisted of 14 patients in stage 0, 20 patients in stage I, 21 patients in stage II, 22 patients in stage III, and 23 patients in stage IV. The stage IV CRC showed a significantly higher rate of positive GroP expression than the other stages (stage 0, 14% (2/12); stage I, 25% (5/20); stage II, 43% (9/21); stage III, 36% (8/22); stage IV, 74% (17/23)) (Figure 1B). Notably, no immunoreactivity to DG2 was observed in normal colorectal tissues. These data indicate that GroP modification is promoted as cancer progresses.

### 2.2. GroP Modification Promotes Cancer Migration

To address the GroP modification function in cancer, we examined the possible effects of the GroP modification enhanced by TagD overexpression on the proliferation and migration of HCT116 cells as a model of colorectal cancer cells. While the CCK-8 assay showed virtually no effect on cell proliferation, the migration ability was enhanced upon TagD overexpression, as demonstrated by wound-healing and transwell assays (Figure 2). These data indicated that enhancement of the GroP modification promoted cancer cell migration, which could be caused by an increase in CDP-Gro amount in cells.

### 2.3. Subsubsection PCYT2 Synthesizes CDP-Gro

We next attempted to identify mammalian enzymes that catalyze the CDP-Gro synthesis, assuming their homology with bacterial TagD. From the human genome database, we found five genes, CDS1, CDS2, PCYT1A, PCYT1B, and PCYT2, that share >30% homology with the TagD gene (Table 1). We produced recombinant proteins encoded by these genes in HCT116 cells and evaluated their CDP-Gro synthetic activity in vitro (Figure 3A). Consequently, we found that only PCYT2 could synthesize CDP-Gro, while the remaining four proteins had no activity. To date, PCYT2 has been characterized as ethanolamine-phosphate cytidylyltransferase, catalyzing the formation of CDP-ethanolamine (CDP-Etn) [35,36]. This enzyme has two major splicing isoforms, PCYT2α and PCYT2β, which were shown to be comparable in terms of the enzymatic activities to CDP-Gro synthase as well as CDP-Etn synthase (Figure 3B and Table 2).

Furthermore, to investigate whether PCYT2 actually synthesizes CDP-Gro in cells, we established PCYT2-KO cells (Appendix A) and quantified intracellular CDP-Gro and CDP-Etn in the cells based on their extraction ion chromatograms (Figure 4A). As expected, CDP-Etn was markedly reduced in the PCYT2-KO cells (Figure 4B). Moreover, CDP-Gro was undetectable in the PCYT2-KO cells. Furthermore, the CDP-Gro level was recovered in the PCYT2-KO cells rescued by overexpressing PCYT2 (Figure 4B). These results indicate that PCYT2 is the enzyme responsible for the synthesis of CDP-Gro in mammalian cells.

### 2.4. GroP Modification of α-DG Critically Depends on PCYT2

We overexpressed the Fc-fused α-DG recombinant protein with amino acid substitution and deletion of α-DG (α-DG373(T322R)-Fc) in the wild-type (WT) and PCYT2-KO cells. We analyzed their GroP modifications by nanospray liquid chromatography–tandem mass spectrometry (LC-MS/MS) (Figure 5). The data demonstrated that the GroP modification of matriglycans was not detected in the PCYT2-KO cells in contrast to the WT cells. In addition, the overexpression of PCYT2 resulted in a significant reduction in the matriglycans on α-DG (Figure 6). These data suggested the critical involvement of PCYT2 in the GroP modification of α-DG by supplying CDP-Gro as a donor substrate.

We detected the expression of PCYT2 in serial sections of stage IV CRC tissues using the anti-PCYT2 antibody. Of 23 stage III CRCs, 13 CRCs revealed positive expression of PCYT2. Interestingly, the immunostaining score was significantly correlated between GroP and PCYT2 expressions in the CRC tissues (r = 0.614, *p* = 0.002) (Figure 7).

## 3. Discussion

The glycan-mediated cell–cell communications provide the basis of the higher functions of our body, which are represented by the immune and nervous systems [37,38]. Therefore, the glycans have to be properly displayed on cell surfaces, and glycosylation abnormalities cause disorders of cellular society, resulting in severe pathological symptoms, as exemplified by cancer malignant transformation [39]. The glycan formation involves a series of glycosyltransferases whose up- and downregulations can control glycan elongation and consequent cellular interactions and, moreover, sophisticate functions [40]. In the case of matriglycan formation on α-DG, at least nine different glycosyltransferases play indispensable roles, and their misfunctions result in CMD. Recently, we found that GroP modification provides an alternative mode of regulation of elongation of the α-DG O-glycans by capping the core M3 structure [20]. This modification is catalyzed by the Janus-like enzyme FKTN, which also transfers Rbo5P and thereby positively contributes to the glycan elongation [21], suggesting some additional mechanism for selectively regulating the modification by GroP as a glycan termination factor.

Our immunohistochemical analysis for GroP modification in human colorectal cancer tissues indicated a significantly positive correlation between colorectal cancer progression and GroP modification. The data are consistent with a reduction in expression levels of matriglycan observed in high-grade tumors in the previous studies [26,30]. Matriglycans are responsible for the formation of basement membranes in tumor invasion through the interaction of ECM proteins such as laminin [41]. For example, an increase in amount of matriglycan by overexpression of LARGE results in suppression of the migration ability of cancer cells [32,33]. In addition, depletion of matriglycans by lowering the expression of B4GAT1 promotes cell migration and thereby increases tumor formation [34]. It has also been pointed out that inadequate expression of the matriglycans activates the integrin-mediated AKT/ERK pathway involved in cancer migration [34]. Because integrins are competitive with α-DG in terms of laminin-binding, the loss of the matriglycans of α-DG enhances the integrin–laminin interactions, thereby enhancing the signaling leading to cancer metastasis.

This study identified PCYT2 as a CDP-Gro synthase and that the GroP modification critically depends on this enzyme. We furthermore found a significantly positive correlation between GroP modification and PCYT2 expression. Moreover, we demonstrate that GroP modification promotes the migration of cancer cells. Finally, we integrate these findings into a cancer metastasis model wherein the GroP modification by PCYT2 suppresses formation of matriglycans and thereby abrogates the glycan-mediated cell adhesion to ECM, thereby promoting migration and invasion of cancer cells.

Lu et al. showed that matriglycans in the breast cancer cell line MCF7 increased upon culturing with the ribitol as a supplement [27]. The enhancement of matriglycan was supposed to result from increased level of CDP- ribitol (CDP-Rbo). The present study demonstrates that overproduction of CDP-Gro causes depletion of matriglycans by capping the non-reducing end of M3 structure through GroP modification. These data suggest that the balance between CDP-Gro and CDP-Rbo in the cells is an important determining factor of the elongation of matriglycans.

Thus, our findings raise the possibility of novel approaches for cancer treatment by antagonizing GroP modification. Particularly, PCYT2 can be a potential target for developing anticancer drugs. However, the present study indicates that PCYT2 is also a dual-function enzyme that synthesizes both CDP-Gro and CDP-Etn, known to be a vital precursor in the biosynthesis of phosphatidylethanolamine [42]. Therefore, it is important to explore how these two enzymatic activities are regulated from an anticancer drug discovery perspective. An alternative approach will be developing therapeutic antibodies targeting the GroP groups displayed on malignant cancer cells.

The present study highlights the pathological aspect of GroP modification in the context of cancer metastasis. Conversely, it is also important to elucidate the physiological roles of this modification. Lymphocytes need to be freed from the adhesion to the ECM for migrating throughout our body. Indeed, T cells have been reported to lose their matriglycans upon maturation [43]. An intriguing possibility is that the PCYT2-mediated GroP modification is involved in such regulations. Further elucidation of the regulatory mechanisms of this modification system will contribute to a deeper understanding of the physiological control of cell adhesion and also to develop therapeutic approaches for a disorder of the system.

While preparing this manuscript, Dr. Endo and his coworkers published a report showing that PCYT2 was a mammalian CDP-Gro synthase [44], consistent with our findings.

## 4. Materials and Methods

### 4.1. Antibodies

The following antibodies were used in this study: anti-Flag (M2) mouse IgG monoclonal antibody (mAb) (Sigma-Aldrich, St. Louis, MO, USA); anti-c-Myc (9E10) mouse IgG mAb (Wako, Osaka, Japan); anti-α-dystroglycan (IIH6C4) mouse IgM mAb (Millipore, Billerica, MA, USA); anti-human dystroglycan (AF6868) sheep IgG polyclonal antibody (pAb) (R&D systems, Minneapolis, MN, USA); anti-PCYT2 (14827) rabbit IgG pAb (Proteintech, Rosemont, IL, USA); anti-β-actin mouse IgG mAb (Sigma-Aldrich); HRP-conjugated anti-mouse IgM mAb (Thermo Fisher Scientific, Waltham, MA, USA); HRP-conjugated anti-mouse IgG mAb (Invitrogen, Carlsbad, CA, USA); HRP-conjugated anti-mouse IgG antibody (Dako, Glostrup, Denmark); HRP-conjugated anti-sheep IgG antibody (Sigma-Aldrich); HRP-conjugated anti-rabbit IgG polyclonal antibody (Cell Signaling, Danvers, MA, USA); mouse monoclonal IgM Ab reactive with GroP-modified α-DG, DG2 [24].

### 4.2. Patients

One hundred colorectal cancer (CRC) patients who underwent surgical and endoscopic resection at Nagoya City University Hospital from March 2018 to December 2019 were enrolled in the present study. The clinical stage was determined based on the final pathological diagnosis after resection, according to the 7th edition of the UICC-TNM classification. The study protocol was approved by the institutional review board (IRB) of Nagoya City University (IRB #. 60-19-0047), and it was conducted following the ethical guidelines of the 1975 Declaration of Helsinki (6th revision, 2008).

### 4.3. Immunohistochemistry

Immunohistochemical staining was performed using serial sections of each sample as follows. All tumor samples were resected from primary CRC tissues, fixed in formalin, and embedded in paraffin. Consecutive sections (4 μm thick) were deparaffinized and dehydrated. After inhibition of endogenous peroxidase activity by immersion in 3% H_2_O_2_/methanol solution, antigen retrieval was achieved by heating the samples in 10 mM citrate buffer (pH 6.0) in a microwave oven for 10 min in at 98 °C. Sections were then incubated with primary antibodies against 20 μg/mL of the anti-GroP DG2 antibody and anti-PCYT2 antibody at dilution of 1:200. After thorough washing, samples were incubated with biotinylated secondary antibodies and then with avidin–biotin horseradish peroxidase complexes. Finally, immune complexes were visualized by incubation in 3,3’-diaminobenzidine tetrachloride. Nuclear counterstaining was accomplished using Mayer’s hematoxylin. All immunostained specimens were assessed by an independent observer blinded to all clinical information. When more than 10% of all cancer cells in each section were stained, the staining intensity score was classified as negative (0), weak (1), moderate (2), or strong (3). Finally, immunostaining scores of (2) and (3) were defined as positive. In the statistical analysis, the binary data were analyzed using the chi-squared test or Fisher’s exact probability test, as appropriate. The nonparametric Spearman’s rank correlation coefficient (r) was used as a correlation measure. All statistics were calculated using SPSS Statistics version 25 (IBM Corp., Tokyo, Japan). A two-tailed *p*-value of less than 0.05 was considered statistically significant.

### 4.4. cDNA Construction

The expression plasmid for the α-DG373(T322R)-Fc was constructed as previously described (20). In this plasmid vector, the coding sequence of human α-DG (1-373) with amino acid substitution of threonine with arginine at position 322 was subcloned into pEF-Fc. Expression vector of Flag-tagged TagD was constructed as previously described (24). In the expression plasmid for Flag-tagged PCYT2 (Uniprot KB: Q99447-1) recombinant protein, the coding sequence of human PCYT2 with flag peptide (DYKDDDDK) purchased from addgene (#81074) was cloned into the pCAG-Neo vector (WAKO). The expression plasmids for Myc-tagged CDS1 (UniProt KB: P98191), CDS2 (UniProt KB: Q99L43), PCYT1A (UniProt KB: P49586), and PCYT1B (UniProt KB: Q811Q9) recombinant proteins were purchased from Origene. In bacterial expression plasmids for His6-tagged PCYT2β (Uniprot KB: Q99447-1) and PCYT2α (Uniprot KB: Q99447-3) recombinant proteins, the coding sequences were cloned into pColdI vector (Takara Bio Inc., Shiga, Japan).

### 4.5. Cell Culture and Transfection

HCT116 (ATCC) cells were maintained in Dulbecco’s Modified Eagle’s Medium (DMEM)-high glucose (Thermo Fisher Scientific) supplemented with 10% fetal bovine serum (FBS) (Cell Culture Technologies, Gravesano Switzerland) for 37 °C, at 5% CO_2_. For cDNA transfection, cells were grown overnight and transfected using polyethyleneimine “Max” (Polysciences, Inc., Warrington, PA, USA), as previously described [20].

### 4.6. Western Blot

Western blots were performed as described previously [20,24]. Recombinant proteins or cell lysates were subjected to sodium dodecyl sulfate–polyacrylamide gel electrophoresis (SDS-PAGE) and subsequently transferred to polyvinylidene difluoride membranes (Merck Millipore, Burlington, MA, USA). After blocking with Blocking One solution (Nacalai Tesque, Tokyo, Japan), the membranes were incubated with primary antibodies, followed by incubation with respective horseradish peroxidase (HRP)-conjugated secondary antibodies. For the detection of α-DG, the lysates were enriched by wheat germ agglutinin (WGA)-agarose (Sigma-Aldrich) and subjected to Western blot analysis. The protein bands were developed with Immobilon Western Chemiluminescent HRP substrate solution (Millipore) and were imaged with an Amersham™ Imager 600 (GE Healthcare, Chicago, IL, USA). After removing the antibodies or avidin by Western Blot Stripping Buffer (Takara Bio Inc.), the membrane was reproved using different antibodies.

### 4.7. Cell Proliferation Assay

The cell counting kit-8 (CCK-8^®^) (Dojindo, Kumamoto, Japan) was used for the cell proliferation assay, according to the manufacturer’s instructions. Briefly, at 24 h after transfection with TagD expression or control vectors, 104 cells per well were seeded in 96-well plates and then incubated at 37 °C. After 0, 24, 48, and 72 h, 10 μL of CCK-8^®^ solution was added to each well and incubated at 37 °C for 2 h. The formazan formation was assessed using a plate reader (Nivo S, PerkinElmer, Waltham, MA, USA) at 450 nm.

### 4.8. Cell Migration Assay

Migratory ability of HCT116 cells was examined by wound-healing and transwell migration assays. A scratch wound assay evaluated cell migration and motility. At 24 h after transfection of TagD expression or control vectors, the transfected cells overgrown in the 33 mm glass-bottom dishes (Iwaki, Tokyo, Japan) were coated with poly-L-lysin. Then, the monolayer cells were scratched by sterile pipette tips uniformly. The cell mobility was monitored for 24 h using a Keyence BZ-X800 microscope (Keyence, Osaka, Japan) equipped with a time-lapse module. The wound closure areas were measured with ImageJ software (1.48 q, Rayne Rasband, National Institutes of Health, Bethesda, MA, USA). Cell migratory ability of wound-healing was assessed using the following formula: [(wound area at 0 h)–(wound area at indicated 24 h)]/(wound area at 0 h). A higher score indicates a better migratory ability.

In transwell assay, cells were placed onto 24-well transwell chambers with a pore size of 8 μm (FALCON). The cells had migrated to the reverse face of the membrane. A total of 24 h after transfection with TagD expression or control vectors, the cells were seeded at 1.5 × 10^5^ cells/well in 200 μL medium (supplemented with 0% FBS) in the upper chamber. At the same time, 500 μL medium (supplemented with 10% FBS) was added in the lower chamber. After incubating for 24 h, the cells on the reverse face were fixed in 4% paraformaldehyde phosphate buffer solution (Wako) and then stained by 1% Crystal Violet (Sigma-Aldrich) and 2% ethanol.

### 4.9. Protein Expression and Purification

Myc-tagged CDS1, CDS2, PCYT1A, and PCYT1B, as well as Flag-tagged PCYT2 and TagD recombinant proteins, were produced by transfecting expression vectors in HCT116 cells. After 48–72 h of transfection, recombinant proteins were purified by anti-c-Myc Antibody Beads (10D11; Wako) or Anti-FLAG M2 affinity gel (Sigma-Aldrich) following the standard protocol. His6-tagged PCYT2α or PCYT2β recombinant protein was produced in *E. coli* BL21-CodonPlus (DE3) (Agilent Technologies, Santa Clara, CA, USA), purified by cOmplete His-Tag purification columns (Roche, Basel, Switzerland), and concentrated to 1 mg/mL using Amicon Ultra 10,000 NMWL (Merck). Recombinant α-DG 373(T322R)-Fc proteins were prepared from the culture supernatants of the transfected cells as described previously [20].

### 4.10. HPLC Analysis of In Vitro Enzymatic Reactions

CDP-Gro synthetic activity was examined using a mixture of 625 µM glycerol-3-phosphate, 300 µM CTP, 1 mM MgCl2, 1 mM DTT, and ~12 µg/mL PCYT2 or one of the other candidate enzymes in 4 mM Tris-HCl (pH 7.5). The mixture was incubated at 37 °C overnight. Enzymatic activities of PCYT2 were quantified using a mixture of varying concentrations of ethanolamine phosphate or glycerol-3-phosphate, 300 µM CTP, 1 mM MgCl2, 1 mM DTT, and 72 µg/mL of PCYT2α or 24 µg/mL of PCYT2β in 4 mM Tris-HCl (pH 7.5). The reaction mixture was incubated at 37 °C for 3 h. After quenching the reaction by boiling at 94 °C for 5 min, the reaction mixture was twofold diluted with acetonitrile and subjected to HPLC analysis as described previously (17). CDP-Gro was measured by UV absorbance at 269 nm. The elution positions of CDP-Gro and CDP-ethanolamine were identified using authentic standards (Sigma-Aldrich).

### 4.11. CRISPR/Cas9 Genome Editing

The pSpCas9(BB)-2A-Puro (PX459) vector (#48139) was used for the cloning via BbsI site and the expression of single target gRNAs to generate the KO cells deficient in PCYT2, with target sequences as follows: PCYT2, 5′-CGTTGTCCTTTTCCTAGAGG-3′ and 5′-GTACAGGTGAGTCTCCACCG-3′. The two constructed pSpCas9(BB)-2A-Puro (PX459) vectors against individual genes were transiently transfected into HCT116 cells in a six-well plate using polyethyleneimine “Max” (Polysciences, Inc.). At 48 h post-transfection, the cells were treated with puromycin (Thermo Fisher Scientific). Knockout cells were selected with 7.5 μg/mL puromycin for 5 days, then cultured in medium without puromycin for 2–3 weeks. The cells were replated as single cells in 6 cm dishes. Deletions of target genes in the clones were confirmed by PCR, sequencing analysis, and Western blotting (Appendix A).

### 4.12. Stable Cell Line Generation

PCYT2-overexpressing cells and PCYT2-KO rescue cells were generated by 600 μg/mL G418 (Thermo Fisher Scientific) selection for 14 days after the transfection with a Flag-tagged PCYT2 vector. The expression of PCYT2 in G418-resistant cells was checked by Western blot using an anti-PCYT2 antibody (Appendix A).

For transposon-mediated gene transfer, cells were transfected with the Tol2 transposase expression vector (pCAGGS-T2TP) and the donor vector (pT2K-CAGGS-rtTA-PCYT2) using PEI-Max. One day after the transfections, cells were selected with 7.5 μg/mL puromycin for 5 days, then cultured in a medium without puromycin for 2 weeks.

### 4.13. LC-MS/MS Analysis of Nucleotide Derivatives

The HILIC-ESI-MS/MS method was used [45,46] to determine the abundances of CDP-glycerol and CDP-ethanolamine in cultured cells. Briefly, cultured cells plated on a 6 cm dish were collected in ice-cold 70% ethanol (1.5 mL) and spiked with an unnatural GDP-glucose (1 nmol). The extracts were centrifuged at 16,000× *g* for 15 min at 4 °C, and the supernatants were lyophilized. As reported previously [46], the samples were dissolved in 2 mL of 10 mM NH_4_HCO_3_ and purified on an Envi-Carb column. The 70% ethanol extraction precipitate was dissolved in 2% SDS to measure the protein concentrations, followed by quantifying protein concentration using Pierce BCA Protein Assay Kit (#23227).

HILIC-ESI-MS/MS was performed on an LCMS-8060 (Shimadzu, Kyoto, Japan) coupled with a Nexera HPLC system (Shimadzu, Kyoto, Japan). Chromatography was performed on a zwitterionic (ZIC) column with phosphocholine phase (ZIC-cHILIC, 2.1 mm i.d. × 150 mm, 3 µm; Merck SeQuant, Sweden) [45]. CDP-glycerol and CDP-ethanolamine were analyzed in the multiple reaction monitoring mode using specific precursor ion [M-H]^−^ and product ions pairs as follows: CDP-glycerol *m*/*z* 476.2 → 322.3; CDP-ethanolamine, *m*/*z* 447.4 → 112.2. The metabolite levels were normalized as pmol/mg proteins.

### 4.14. LC-MS/MS Analysis of Glycopeptides

Glycopeptides derived from α-DG were identified by LC-MSMS analyses as previously described [20,24]. Briefly, the Fc-fused α-DG [α-DG 373(T322R)-Fc] were expressed in HCT116 and the PCYT2-KO cells. After purification of the recombinant proteins followed by separation using SDS-PAGE, the gel bands containing α-DG373(T322R)-Fc were excised and subjected to reduction, alkylation, and then in-gel digestion by trypsin (Promega, Madison, WI, USA). Digested peptides purified using a ZipTip C18 (Merck Millipore) were analyzed by LC-MS/MS on an Orbitrap Fusion Tribrid (Thermo Fisher Scientific) coupled to an Easy-nLC 1200 (Thermo Fisher Scientific). The identities of glycopeptides were then manually verified based on positive detection of the expected peptide pyrQIHATPTPVR b and y ions, and/or the peptide core ion at *m/z* 551.804 (z = 2) or 1102.600 (z = 1) in their respective HCD MS/MS spectra, as described previously [20,24].

## Figures and Tables

**Figure 1 ijms-23-06662-f001:**
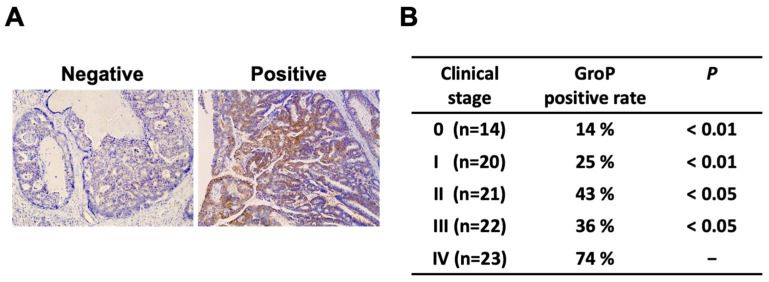
Immunohistochemical analysis for glycerol phosphate (GroP) modification in human colorectal cancer tissues. (**A**) Representative images of staining with DG2. The left and right panels represent negative and positive DG2 staining, respectively, in human colorectal cancer tissues (×100). (**B**) GroP expression according to disease stage. The *p*-value in each stage was calculated by comparison with the positive rate in stage IV.

**Figure 2 ijms-23-06662-f002:**
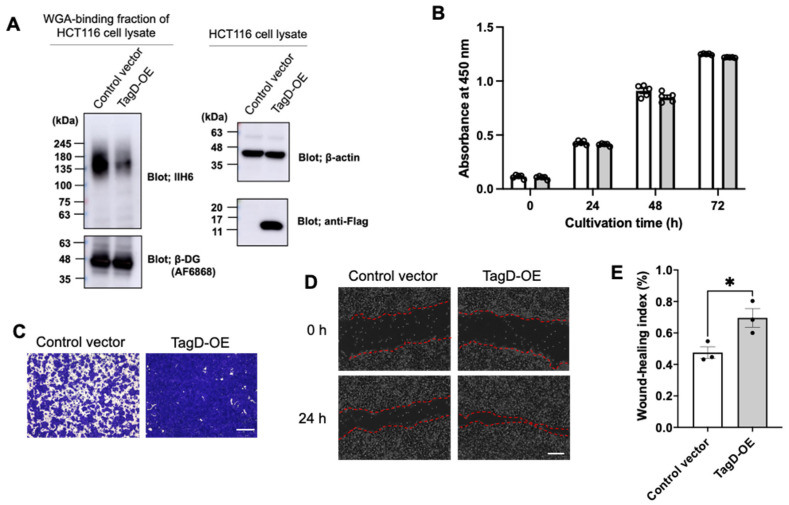
Effects of enhancement of the GroP modification on cancer cell behaviors. (**A**) TagD overexpression reduced the expression of matriglycans. Cell lysates containing equal amounts of total proteins prepared from HCT116 cells transfected with TagD-Flag-expressing vector or control vector were subjected to immunoblot analysis using IIH6 and AF6868 along with anti-β-actin and anti-Flag antibodies. (**B**) Proliferation assay of HCT116 cells transfected with TagD-expressing vector or control vector at 0, 24, 48, and 72 h. Error bars represent the standard error of the mean (SEM) (*n* = 3). (**C**) Transwell assay (scale bar = 600 μm) and (**D**) wound-healing assay (scale bar = 200 μm) for HCT116 cells at 24 h after transfection with TagD-expressing or control vector. (**E**) Wound-healing index of TagD-overexpressing cells and control cells. Individual data points are represented by black dots on the bar graph. Error bars represent the SEM (*n* = 3). Significant differences (*) were calculated compared with WT index using two-tailed unpaired Student’s *t*-test (*p*  <  0.05).

**Figure 3 ijms-23-06662-f003:**
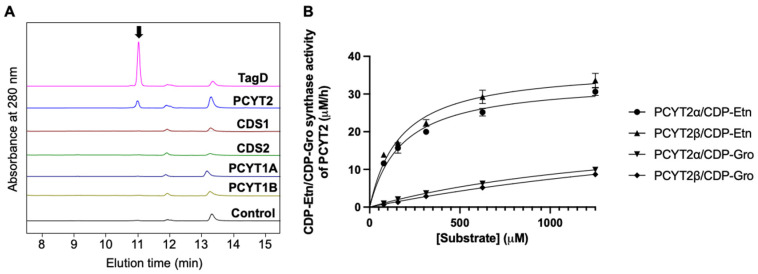
Characterization of enzymatic activities of PCYT2. (**A**) HPLC profiles of the reaction mixtures in the presence and absence of the candidate enzymes listed in Table 1 and positive (TagD) and negative controls. The peak corresponding to CDP-Gro is indicated with the arrow. (**B**) The enzymatic activities of PCYT2α and PCYT2β with various glycerol-3-phosphate or ethanolamine phosphate and CTP concentrations are plotted. CDP-Gro/CDP-Etn synthase activities of PCYT2α (▼/●) and PCYT2β (Ø◆/▲) are represented, respectively. The data were fitted to the Michaelis–Menten equation by nonlinear regression (GraphPad Prism 9; https://www.graphpad.com/scientific-software/prism/). The apparent Michaelis constant (*K*_m_) and the maximal velocity (*V*_max_) were calculated.

**Figure 4 ijms-23-06662-f004:**
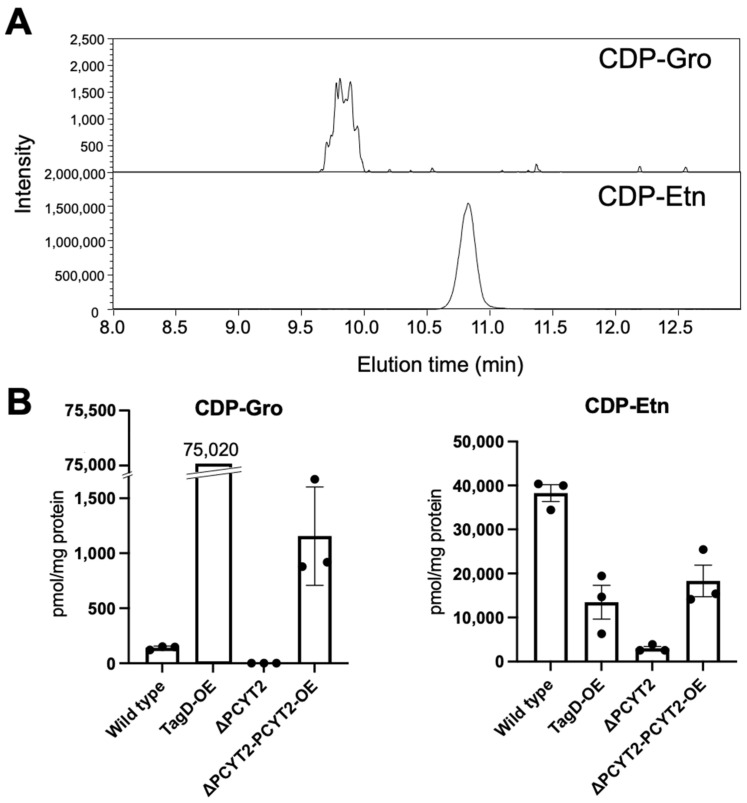
LC-ESI-MS/MS analysis of CDP-Gro and CDP-Etn from HCT116 cells. (**A**) Representative extraction ion chromatograms of CDP-Gro (upper panel) and CDP-Etn (lower panel) derived from HCT116. (**B**) HCT116 wild-type cells, TagD-overexpressing cells (TagD-OE), PCYT2-KO (ΔPCYT2), and PCYT2-KO rescue cells (ΔPCYT2-PCYT2-OE) were analyzed, and the data were normalized to units of pmol per milligram of protein. Individual data points are represented by black dots on the bar graph. Error bars represent the standard error of the mean (SEM) (*n* = 3).

**Figure 5 ijms-23-06662-f005:**
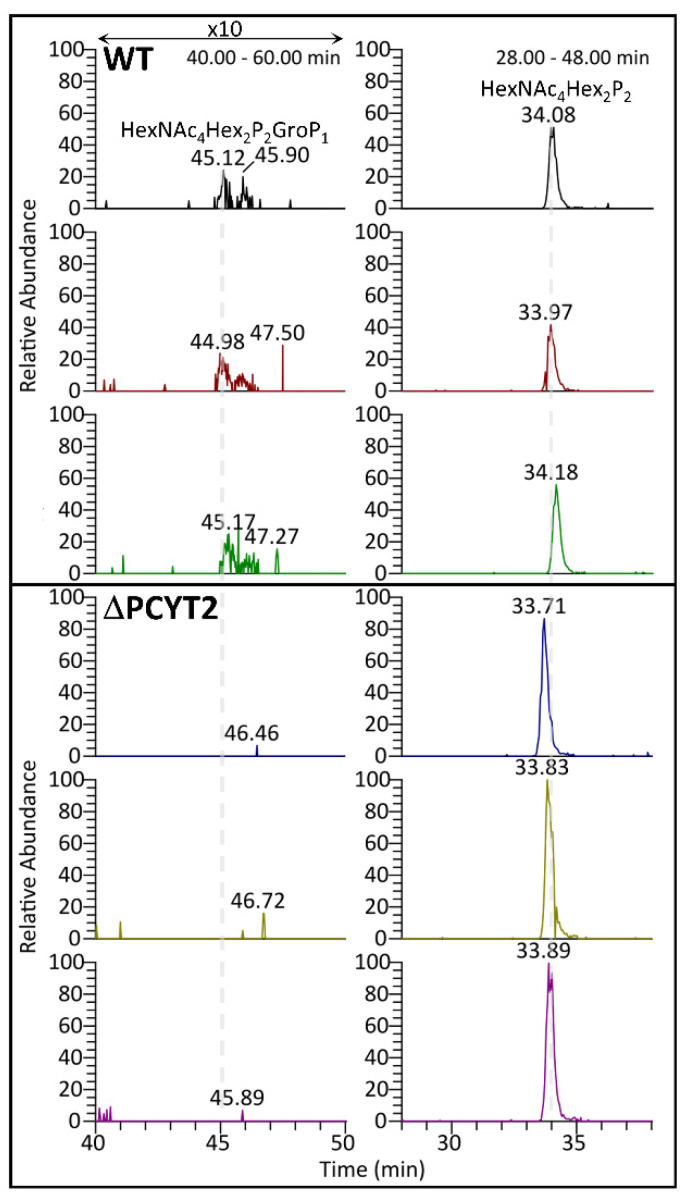
Glycan modification on α-DG373(T322R)-Fc expressed in HCT116 wild-type and PCYT2-KO cells. The extracted ion chromatograms of selected ^313^pyrQIHATPTPVR^322^ on α-DG carrying phosphorylated core M3 glycoforms, HexNAc_4_Hex_2_P_2_GroP_1_ (**left**) and HexNAc_4_Hex_2_P_2_ (**right**), in biological triplicate of WT (upper 3 panels) and ΔPCYT2 (lower 3 panels) derived from HCT116. The HexNAc_4_Hex_2_P_2_GroP_1_ glycoforms were only detected in WT, but not ΔPCYT2, in contrast with the detections of phosphorylated core M3 structures, HexNAc_4_Hex_2_P_2_GroP_1_.

**Figure 6 ijms-23-06662-f006:**
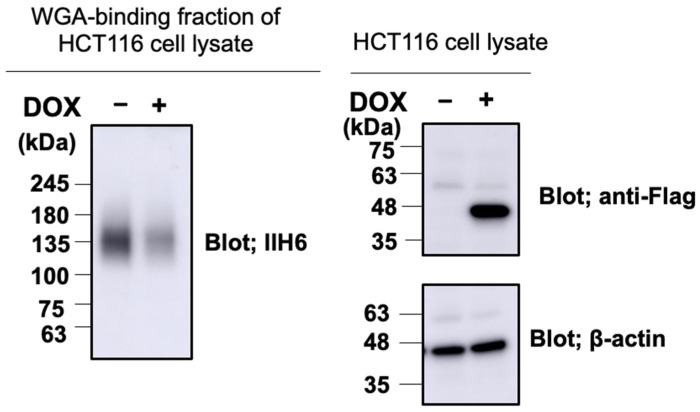
The expression of matriglycans was reduced by PCYT2-overexpression. HCT116 clone stably transfected with the doxycycline (Dox)-inducible PCYT2 construct; −Dox, uninduced cells; + Dox, induced cells. Cell lysates containing equal amount of total proteins prepared from HCT116 cells with (+) or without (−) Dox were subjected to immunoblot analysis using anti-Flag antibody and anti-β-actin, or to WGA enrichment followed by immunoblot analysis using IIH6.

**Figure 7 ijms-23-06662-f007:**
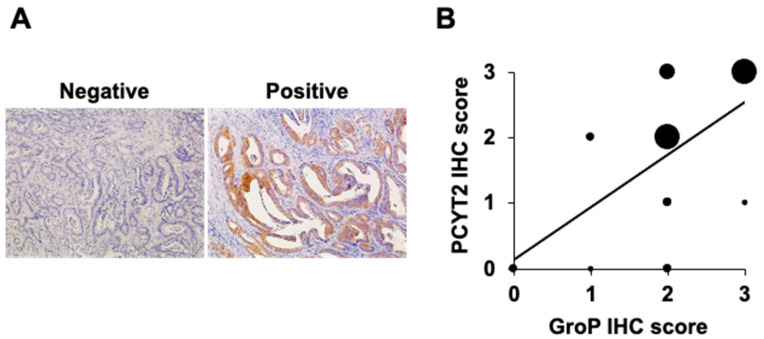
Correlation between GroP modification and PCYT2 expression levels in colorectal cancer tissues. (**A**) Representative images of PCYT2 staining. The left and right panels represent negative PCYT2 and positive PCYT2 staining, respectively, in human colorectal cancer tissues (×100). (**B**) Mapping of correlation between GroP modification and PCYT2 expression levels based on the staining scores. Staining scores of both GroP and PCYT2 were calculated at stage IV colorectal cancer tissues, and data were analyzed using the Spearman rank correlation. Dot size corresponds to the number of colorectal cancer tissue samples.

**Table 1 ijms-23-06662-t001:** Sequence identities of homologs genes with the TagD gene calculated utilizing the BLAST 2.0 algorithm.

Gene	Protein	Identity	Query Cover
CDS1	Phosphatidate cytidylyltransferase 1	57%	10%
CDS2	Phosphatidate cytidylyltransferase 2	67%	8%
PCYT1A	Choline-phosphate cytidylyltransferase A	35%	93%
PCYT1B	Choline-phosphate cytidylyltransferase B	36%	93%
PCYT2	Ethanolamine-phosphate cytidylyltransferase	30%	89%

**Table 2 ijms-23-06662-t002:** The enzymatic parameters of CDP-Gro or CDP-Etn synthase activities of PCYT2 isoforms using glycerol-3-phosphate or ethanolamine phosphate as substrates, respectively.

	CDP-Etn		CDP-Gro	
	PCYT2α	PCYT2β	PCYT2α	PCYT2β
*V*_max_ (mM/h)	33.7	37.3	22.7	28.9
*K*_m_ (mM)	181.2	171.2	1614	2899
Catalytic efficiency (*k*_cat_/*K*_m_) (mM^−1^h^−1^)	0.3530	0.3974	0.0089	0.0060

## Data Availability

The data presented in this study are available on request from the corresponding authors. The mass spectrometric data of LC-MS have been deposited to the Mass Spectrometry Interactive Virtual Environment (MassIVE) with the dataset ID: MSV000089635.

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
