# Peer review of "Cancer Malignancy Is Correlated with Upregulation of PCYT2-Mediated Glycerol Phosphate Modification of α-Dystroglycan"

_ijms, 2022, doi:10.3390/ijms23126662_

Round 1
Reviewer 1 Report
Dear Authors,
Congratulations on your work put into this interesting manuscript. However, before acceptance, I suggest to be revised Introduction, Material and methods and Discussion sections. Precisely, please expand Introduction and Discussion sections with more recent studies (2018-2022) if possible. These sections are crucial, but in the current are quite brief. At Western blot experiment, please offer a short description of the protocol. Overall, good job.
Kind regards,
The Reviewer
Author Response
We thank the reviewer for the constructive comments. As per the reviewer’s comment, we expanded the introduction and discussion sections with references of recent papers (p.2, lines 68-77 and p.9, lines 235-248 and 256-262). We also provided a detailed description of the western blot in the materials and method section in the revised manuscript (p.11, lines 352-363).
Reviewer 2 Report
Authors presented some interesting findings about the role of the glycerol phosphate modification in cancer malignancy. The results of this study could also suggest possible novel approaches for cancer treating in the future. There are just some minor improvements that authors might consider:
1. The results are scientifically strong and well presented. However, the aim of the study should be better defined (lines 70-72). It seems that a summary of selected findings is included in this section instead of the aim of the study.
2. The Discussion should be improved. All the important findings are summarized in Discussion section, but they are poorly discussed.
3. A cancer metastasis model is mentioned in discussion (lines 230-232). I would recommend that authors describe this model in more details.
Author Response
- The results are scientifically strong and well presented. However, the aim of the study should be better defined (lines 70-72). It seems that a summary of selected findings is included in this section instead of the aim of the study.
We thank the reviewer for the insightful comment. Accordingly, we modified the Introduction for clearly the purpose of this study in the revised manuscript (p.2, lines 68-79).
- The Discussion should be improved. All the important findings are summarized in Discussion section, but they are poorly discussed.
We concur with the reviewer’s remark and expanded the discussion section in the revised manuscript p.9, lines 235-248 and 256-262).
- A cancer metastasis model is mentioned in discussion (lines 230-232). I would recommend that authors describe this model in more details.
As per the reviewer’s comment, the manuscript has been modified to describe the model in more detail (p.9, lines 235-255).